# Older Adults’ Attitudes Regarding COVID-19 and Associated Infection Control Measures in Shanghai and Impact on Well-Being

**DOI:** 10.3390/jcm11247275

**Published:** 2022-12-08

**Authors:** Zhimin Xu, Lixian Cui, Gabriela Lima de Melo Ghisi, Xia Liu, Sherry L. Grace

**Affiliations:** 1Cardiology Department, Xinhua Hospital Affiliated to Shanghai Jiao Tong University School of Medicine, Shanghai 200092, China; 2Division of Arts and Sciences, New York University Shanghai, Shanghai 200122, China; 3Cardiovascular Prevention and Rehabilitation Program, Toronto Rehabilitation Institute, University Health Network, University of Toronto, Toronto, ON M4G 2R6, Canada; 4Chengdu Wanda UPMC Hospital, Chengdu 610218, China; 5Faculty of Health, York University, Toronto, ON M3J 1P3, Canada; 6KITE-Toronto Rehabilitation Institute & Cardiac Rehabilitation Research, Peter Munk Cardiac Centre, University Health Network, University of Toronto, Toronto, ON M5G 2C4, Canada

**Keywords:** COVID-19, psychological distress, older Chinese adults, attitude, surveys and questionnaires

## Abstract

This cross-sectional study investigated health management, well-being, and pandemic-related perspectives in Shanghainese adults ≥50 years at the early stages of COVID-19 using a self-report survey in March–April, 2020. Items from the SSS, PHQ-9 and GAD-7 were administered. A total of 1181 primarily married, retired females participated. Many participants had hypertension (44.1%), coronary artery disease (CAD; 17.8%), and diabetes (14.5%). While most (n = 868, 73.5%) were strictly following control measures and perceived they could tolerate >6 months (n = 555, 47.0%) and were optimistic (n = 969, 82.0%). A total of 52 (8.2%) of those with any condition and 19 (3.5%) of those without a condition reported that the pandemic was impacting their health. Somatic symptoms were high (29.4 ± 7.1/36), with sleep/cognitive symptoms highest. Totals of 20.2% and 17.0% of respondents had elevated depressive and anxious symptoms, respectively; greater distress was associated with lower income (*p* = 0.018), having hypertension (*p* = 0.001) and CAD (*p* < 0.001), negative perceptions of global COVID-19 control (*p* = 0.004), COVID-19 spread (*p* < 0.001), impact on life/health (*p* < 0.001), compliance with control measures (*p* < 0.001), and the toleration of shorter time control measures (*p* < 0.001) in adjusted analyses. In conclusion, during the initial COVID-19 outbreak, most older adults were optimistic/resilient regarding the epidemic and control measures. However, the distress of older adults was not trivial, particularly in those with health issues.

## 1. Introduction

The coronavirus disease (COVID-19) pandemic has resulted in major negative impacts for economies, health systems, and citizens worldwide. The impact of COVID-19 on the health of all people has been of major concern, particularly before the availability of vaccines or any treatment [1]. Moreover, it has been a particular concern for those in dense urban areas where spread could be greater, as well as those at higher risk of complications. This includes older adults; for instance, hospitalization (45%) and mortality rates (80%) are much higher in individuals ≥65 years [2], as they often suffer from chronic conditions, many of which are themselves associated with increased risk [3].

Measures to control infection spread, such as physical distancing, often necessitate the closure of essential businesses, such that individuals have well-founded anxiety about accessing food and medication. Similarly, stay-at-home orders result in no access to places to be active or connect with others [4]. There has been some study of the impacts of the pandemic and these associated control measures on the psychosocial well-being of older adults, such as anxiety and loneliness; results suggest variability in these indicators, with certain subgroups at greater risk [5,6,7,8,9,10,11,12]. This might be particularly true in China, where filial piety is culturally integral, and control measures could impact this factor. Contrarily, there have also been some findings of a “well-being paradox”: at the beginning of the pandemic, communities rallied to support one another (and particularly older and vulnerable adults) and hence sense of well-being was bolstered [5]. It is also shown that older adults often display less psychological distress than their younger counterparts [13].

Government policy varies worldwide in terms of the implementation of control measures. Based on the COVID-19 stringency index, such strategies are among the most stringent in China [14]. Therefore, the objective of this study was to assess the impact of the pandemic on older adults in the populous city of Shanghai during the first wave when most cases were extant in China, there was much uncertainty, and no vaccine or treatments were available. Impacts on: (1) psychosocial well-being and (2) perceptions related to the pandemic and its control were investigated. Vulnerability and protective factors for elevated depressive and anxiety symptoms were also explored.

## 2. Materials and Methods

### 2.1. Design and Procedure

The voluntary and anonymous online survey was created by the Cardiac Rehabilitation Group of the Health Risk Assessment and Control Committee of the Chinese Preventive Medical Association in Mandarin. Data collection for this cross-sectional study was undertaken between March and April in 2020. The survey was disseminated via the web-based survey platform Wenjuanxing through WeChat by the Community Doctor Group, involving 36 community doctors from all the 16 districts in Shanghai, which belongs to the Shanghai Association of Chinese Integrative Medicine. 

### 2.2. Setting and Participants

Participants of this study were adults ≥50 years old residing in a community for seniors in Shanghai. Exclusion criteria were the following: severe cognitive impairments or any conditions that prevented respondents from being able to understand and voluntarily agree to participate in the research.

At the time of the study, during the initial December 2019 outbreak in Wuhan, COVID-19 was well-controlled in China and a lockdown had been implemented in some cities. Standard precautions were enforced in Shanghai, including wearing masks and physical distancing. For example, policies enforced included no gatherings of more than 10 people (big events, such as wedding and birthday parties, were postponed), not being able to leave Shanghai with visitations being restricted, non-residents of the community not allowed to enter, and some specialist outpatient services were cancelled.

### 2.3. Measures

All items were self-reported. Non-psychometrically validated items were generated by the Cardiac Rehabilitation group (see above), and respondents were asked to respond based on their experience in the prior two weeks. 

The Patient Health Questionnaire-9 (PHQ-9) was administered [15]. The PHQ-9 assesses depression symptom severity in alignment with diagnostic criteria for major depressive disorders. For each item, the patient is asked to rate symptoms over the last 2 weeks, with each item rated on a 4-point Likert scale from 0 (“not at all”) to 3 (“almost every day”). Total scale scores range from 0 to 27, with higher scores indicating a greater severity of depression. The cut-off points are 5, 10, and 15, indicating mild, moderate, and severe depression, respectively. PHQ-9 is considered valid and reliable among Chinese people [16]. 

The Generalized Anxiety Disorder (GAD-7) questionnaire was administered to evaluate symptom of anxiety. It comprises seven items, with each item scored on a 4-point Likert scale ranging from 0 (“not at all”) to 3 (“almost every day”). Total scale scores range from 0 to 21, with higher scores indicating greater anxiety. Cut-off points of 5, 10, and 15 represent mild, moderate, and severe levels of anxiety, respectively [17]. The GAD-7 has been proven to have good reliability and validity in Chinese populations [18].

The Chinese version of the Somatic Self-rating Scale (SSS) was also administered to assess physical (i.e., half the items query each body system), depressive, and anxiety symptoms as well as sleep and cognitive issues [19]. Thus, the SSS is used to measure not only somatic symptoms but also psychosocial well-being. It consists of 20 items categorized in four domains, with each item is rated on a 4-point scale. The total score ranges from 20 to 80, with scores ranging from 20 to 29, 30–39, 40–59, and ≥60 correspond to a normal, mild, moderate, and severe somatic symptom disorder, respectively [20]. The SSS has been proven to have good psychometric properties, including test–retest reliability of 0.9 and a Cronbach’s α of 0.89. In addition, the correlation coefficient between the total score of SSS and the self-rating anxiety scale (SAS) and the self-rating depression scale (SDS) was 0.80 and 0.74, respectively [19].

### 2.4. Statistical Analyses

All data analysis was performed using IBM SPSS statistics version 25.0, with *p* < 0.05 considered statistically significant. Descriptive analysis was first performed. The associations of physical and mental health with perceived pandemic impact on their health were tested using chi-square and *t*-tests, as applicable. 

The association of pandemic-related attitudes and health impact with depressive and anxious symptoms were assessed using *t*-test and chi-square, as applicable. Finally, the multivariate analysis of variance assessing the significant correlates of depressive and anxious symptoms based on the aforementioned analyses as well as associated sociodemographic and clinical correlates were tested using the multivariate analysis of variance (MANOVA).

## 3. Results

Of the 1181 responding participants, most were female, over 65 years, married and living with their partner only, had moderate education, and were retired with the associated healthcare coverage (Table 1). As shown in Table 2, almost half of participants had hypertension, almost one-in-five had coronary artery disease and another 15% had diabetes, while 546 (46.2%) had no assessed condition. Sixty-two (5%) participants reported a psychiatric diagnosis. 

### Impact of Pandemic

Over two-thirds of participants perceived that the impact of the pandemic on their life would be temporary (Table 3). Almost three-quarters strictly complied with COVID-19 prevention and control measures. Almost half of participants reported they could tolerate strict control measures for a long time, with over 20% reporting they could endure the situation for 6 months or 3 months. Not considering those participants who were living with their children, most were isolated from their children, receiving no visits from them. Almost three-quarters transitioned to exercising in their home and only 8.2% reported that the pandemic was having an impact on their health.

Psychosocial well-being early during the pandemic was compromised (Table 2). Mean SSS scores revealed that approximately 42.3% of participants had elevated symptoms and impacts on sleep and cognition were mild. Approximately 20% of participants had elevated depressive symptoms and the same had elevated anxiety.

As shown in Table 2, participants with hypertension, coronary artery disease, cerebrovascular disease, and heart failure perceived that the pandemic had an impact on their health, whereas participants without these conditions did not. Those recognizing an impact of the pandemic on their health condition also were significantly more likely to have a history of psychiatric disorders and correspondingly had significantly greater somatic symptoms (all subscales) as well as depressive and anxiety symptoms. 

As shown in Table 3, participants with elevated depressive symptoms were more worried about future waves and that the pandemic was not temporary and were more likely to barely go out, to be unable to endure strict control measures over prolonged periods, and perceived a significantly greater impact of the pandemic on their health when compared to participants with subclinical symptoms. Participants with elevated anxiety symptoms were significantly less optimistic about global control of the pandemic, were more worried about future waves due to incomers and the economic impact globally of the pandemic, were less likely to view the pandemic spread and impacts as temporary, were less likely to go out, and reported they could endure strict control measures for a shorter time than participants with subclinical symptoms (Table 3). 

Table 4 demonstrates that, in an adjusted MANOVA model, the following correlates were associated with greater depressive and anxiety symptoms: lower income; having hypertension and coronary artery disease; negative perceptions regarding global COVID-19 control; negative perceptions regarding the spread of COVID-19 and the impact of pandemic on their life; greater compliance with prevention and control measures; perceiving negative impact of the pandemic on health conditions; and being less able to tolerate strict COVID-19 control measures over time.

## 4. Discussion

This study examined the impact of the COVID-19 pandemic and its associated control measures in a large sample of older adults, over half of whom had a chronic condition. It was undertaken in one of the most populous cities in the world, in the country with the most stringent control measures globally [21], early in the pandemic when uncertainty was high and there were no known treatments or vaccines. While some studies have examined this in the context of “Western” cultures, where older adults are more isolated and may live in retirement facilities, the context in China is different, with most older adults living with their spouses. Nevertheless, consistent with findings in other countries [22,23,24,25], results show elevated distress, particularly when compared to the general population or medical outpatients more broadly (e.g., 5–10% depressive symptoms) [26].

Indeed, while approximately one-quarter of older adults had elevated depressive symptoms and one-fifth anxiety and most were only going out for food or medicine and limiting visits with children, there was still much optimism and faith that the impacts were temporary and resignation regarding the toleration of long-term control measures (although this should be considered a proxy for their tolerance and motivation at the time of the study). According to Jiang et al. (2020) [6], older Chinese adults were less worried about COVID-19 compared to younger Chinese adults during the initial outbreak; however, as the disease spread, their worries gradually increased. Studies by Tian et al. (2020) [9], Wong et al. (2020) [10], and Zhang et al. (2020) [11] reiterate the negative psychological impact of COVID-19 on Chinese adults.

Study results are particularly concerning given the link between psychosocial well-being and chronic disease onset and outcomes [27]. There are effective treatments for sleep issues [28], depression [29], and anxiety [30], although means of bolstering support are less well-established [31]. Research into the internet-based delivery of evidence-based therapies has been increasing and supportive, although some older adults do not have access to the technology or the digital literacy to take advantage of this COVID-19-friendly modality. Effective pharmacotherapy does not exist for all these conditions, and, in particular, sleep and anti-anxiety medications can be more hazardous in older adults [32].

There are some directions for future research stemming from this study. Whether patients or close contacts had COVID-19 during the period of study was not considered, nor whether participants had to quarantine—more research is needed regarding these factors. Other studies have shown the impact on mobility [33], which was not considered herein. Exercise was assessed in this study, with most older adults reporting they were exercising at home; however, given more objective reports [34], it is likely that their activity was insufficient for optimal primary or secondary prevention [35], thus ultimately leading to likely mobility issues. Moreover, research on the impacts of the pandemic and control measures over time is warranted in order to determine how older adults cope in the longer-term [36]. 

Caution is warranted when interpreting these results. First, this is a cross-sectional study, so the design precludes causal or directional determinations, and we cannot test how psychosocial well-being differs from pre-pandemic levels. Second, the results may not be generalizable beyond China, with its specific political and cultural context, and the city of Shanghai more specifically, where the stringency of control measures was high. Nevertheless, the population of Shanghai is very large, and findings can inform other governments when considering their control measures. Moreover, the similarity of the sample to the larger population is not known given the recruitment strategy, designed to quickly secure data during the lockdown, where full reach and non-responders were not collated. Third, all data were self-reported, and hence socially desirable responding or other measurement errors may have been present, and conditions were not verified through a structured clinical interview or medical charts. Fourth, multiple comparisons were performed, inflating potential type 1 errors.

In conclusion, in this sample of older adults in Shanghai at the beginning of the pandemic, half of whom had a chronic condition and were strictly following prevention and control measures, while optimism was high, levels of somatic symptoms as well as depressive and anxiety symptoms were higher than normative levels. Ensuring access to evidence-based treatment via technology in those who have the digital literacy would have been key at the beginning of the pandemic, knowing now what we do about its duration. More research on the longer-term physical and psychosocial impacts of COVID-19 and the associated prevention and control measures on older adults is needed (Figure 1).

## Figures and Tables

**Figure 1 jcm-11-07275-f001:**
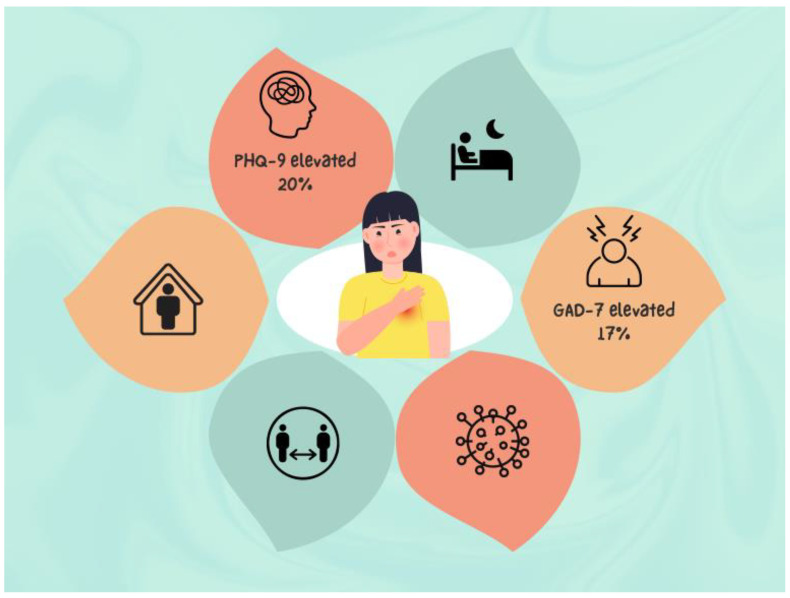
Early in the COVID-19 pandemic in China, older adults were optimistic and highly compliant with prevention and control.

**Table 1 jcm-11-07275-t001:** Participant Sociodemographic Characteristics, N = 1181.

Characteristic	n (%)
Sex (n, % female)	777 (65.8)
*Age (years)*	
50–54	117 (9.9)
55–64	403 (34.1)
65–74	517 (43.8)
75–84	124 (10.5)
>85	20 (1.7)
*Marital Status*	
Married	1042 (88.2)
Widowed	120 (10.2)
Unmarried	19 (1.6)
*Living Situation*	
Alone	98 (8.3)
With partner only	757 (64.1)
With adult children	179 (15.2)
With extended family	147 (12.4)
*Highest Educational Attainment*	
Bachelor and above	347 (29.4)
Junior college or senior high school	662 (56.1)
Junior high school and below	172 (14.6)
*Income Status*	
Retirement pension	1008 (85.4)
No fixed income (e.g., rely on children, state subsidies)	22 (1.9)
Extra money on top of pension (e.g., financial product, rent)	151 (12.8)
*Current or Previous Occupation*	
City worker	308 (26.1)
Farmer	10 (0.8)
Teacher	212 (18.0)
Government	78 (6.6)
Health care provider	199 (16.9)
Veterans	12 (1.0)
Management (private or public)	242 (20.5)
Research	32 (2.7)
Other (civil servants)	88 (7.5)
*Healthcare Insurance*	
Retiree	989 (83.7)
Town residents	139 (11.8)
Off-site	48 (4.1)
Self-pay	5 (0.4)

**Table 2 jcm-11-07275-t002:** Participant Physical and Mental Health and Association with Perceived Pandemic Impact on Health, N = 1181.

	n (%)/ Mean ± SD	Perceived Impact of Pandemic on Health Condition			
		No Effect (n = 1110)	Effect (n = 71)	X^2^/t	*p*
*Chronic Condition*					
Hypertension	521 (44.1)	480 (43.2)	41 (57.7)	5.694	0.017
Coronary artery disease	210 (17.8)	189 (17.0)	21 (29.6)	7.190	0.007
Diabetes	171 (14.5)	156 (14.1)	15 (21.1)	2.696	0.101
Cerebrovascular disease	83 (7.0)	68 (6.1)	15 (21.1)	22.981	<0.001
Renal disease	29 (2.5)	27 (2.4)	2 (2.8)	0.041	0.839
Heart failure	44 (3.7)	32 (2.9)	12 (16.9)	32.758	<0.001
None of the above	546 (46.2)	527 (47.5)	19 (26.8)	11.522	0.001
One or more of above chronic conditions	635 (53.8)	583 (52.5)	52 (73.2)		
*Mental Health*					
History psychiatric disorder (n, % yes)	62 (5.2)	52 (4.7)	10 (14.1)	10.039	0.002
Taking psychoactive medication (n, % yes)	61 (5.2)	51 (4.6)	10 (14.1)	10.408	0.001
Major Life Event (n, % yes)	133 (11.3)	122 (11.0)	11 (15.5)	1.353	0.245
Somatic Self-rating Scale Total (/80)	29.40 ± 7.11	29.01 ± 6.71	35.58 ± 9.85	5.540	<0.001
Somatic Symptoms (/36)	13.32 ± 3.35	13.17 ± 3.24	15.72 ± 4.11	5.117	<0.001
Anxiety symptoms (/20)	6.89 ± 1.90	6.79 ± 1.78	8.51 ± 2.82	5.073	<0.001
Depressive symptoms (/16)	5.85 ± 1.77	5.76 ±1.67	7.30 ± 2.55	5.019	<0.001
Sleep and cognitive symptoms (/8)	3.33 ± 1.09	3.29 ± 1.04	4.06 ± 1.48	4.299	<0.001
SSS normal (20–29)	681(57.7)	658 (59.3)	23 (32.4)	37.848	<0.001
SSS mild (30–39)	409 (34.6)	381 (34.3)	28 (39.4)		
SSS moderate (40–59)	87 (7.4)	68 (6.1)	19 (26.8)		
SSS severe (≥60)	4 (0.3)	3 (0.3)	1 (1.4)		
SSS elevated (>36)	185 (15.7)	154 (13.9)	31 (43.7)	44.821	<0.001
Depressive Symptoms (PHQ-9)	2.50 ± 3.15	2.29 ± 2.81	5.79 ± 5.52	5.295	<0.001
Elevated (5–27)	238 (20.2)	203 (18.3)	35 (49.3)	39.873	<0.001
Sub-clinical (0–4)	943 (79.8)	907 (81.7)	36 (50.7)		
Anxiety Symptoms (GAD-7)	1.88 ± 3.04	1.68 ± 2.71	4.94 ± 5.42	5.033	<0.001
Elevated (5–21)	201 (17.0)	168 (15.1)	33 (46.5)	46.421	<0.001
Sub-clinical (0–4)	980 (83.0)	942 (84.9)	38 (53.5)		

SD, standard deviation; GAD-7, Generalized Anxiety Disorder; PHQ-9, The Patient Health Questionnaire-9; SSS, Somatic Self-rating Scale.

**Table 3 jcm-11-07275-t003:** Pandemic Attitudes and Health Impact and their Associations with Depressive and Anxious Symptoms, N = 1181.

	n (%)	Association with Depressive Symptoms				Association with Anxiety Symptoms			
		Elevated (n = 238)	Subclinical (n = 943)	X^2^/t	*p*	Elevated (n = 201)	Subclinical (n = 980)	X^2^/t	*p*
*Source of Pandemic-related Information*									
Television and newspaper only (traditional media)	86 (7.3)	18 (7.6)	68 (7.2)	0.235	0.954	17 (8.5)	69 (7.0)	2.741	0.220
Traditional and new media (e.g., WeChat)	1090 (92.3)	219 (92.0)	871 (92.4)			182 (90.5)	908 (92.7)		
No media; only family or neighbors	5 (0.4)	1 (0.4)	4 (0.4)			2 (1.0)	3 (0.3)		
*Perspective on Global COVID-19 Control*									
Do not worry about control, as the pandemic will gradually dissipate	142 (12.0)	26 (10.9)	116 (12.3)	6.905	0.075	23 (11.4)	119 (12.1)	9.763	0.021
Optimistic if continue internal controls and limit incomers	969 (82.0)	190 (79.8)	779 (82.6)			158 (78.6)	811 (82.8)		
Once control measures eased, COVID-19 will rebound, impacting lives	30 (2.5)	11 (4.6)	19 (2.0)			6 (3.0)	24 (2.4)		
Regardless of control measures, there will be another COVID-19 wave	40 (3.4)	11 (4.6)	29 (3.1)			14 (7.0)	26 (2.7)		
*Perceptions Regarding COVID-19 Spread in the World*									
Concerned about relatives and friends abroad	173 (14.6)	37 (15.5)	136 (14.4)	0.192	0.661	38 (18.9)	135 (13.8)	3.511	0.061
Glad China has tried to keep the outbreak under control	621 (52.6)	123 (51.7)	498 (52.8)	0.097	0.755	102 (50.7)	519 (53.0)	0.328	0.567
Worried soon second wave due to incomers	220 (18.6)	61 (25.6)	159 (16.9)	9.641	0.002	60 (29.9)	160 (16.3)	20.126	<0.001
Pandemic has spread globally and affected economy	139 (11.8)	33 (13.9)	106 (11.2)	1.261	0.261	41 (20.4)	98 (10.0)	17.366	<0.001
Temporary issue; countries will be united in efforts so there is no need to be dejected	646 (64.7)	95 (39.9)	551 (58.4)	26.289	<0.001	79 (39.3)	567 (57.9)	23.171	<0.001
Only China is my concern; not our business	49 (4.1)	14 (5.9)	35 (3.7)	2.252	0.133	10 (5.0)	39 (4.0)	0.416	0.519
*Impact of Pandemic on Life*									
None	152 (12.9)	20 (8.4)	132 (14.0)	47.145	<0.001	9 (4.5)	143 (14.6)	99.338	<0.001
Only physical distancing	134 (11.3)	29 (12.2)	105 (11.1)			27 (13.4)	107 (10.9)		
Temporary	821 (69.5)	152 (63.9)	669 (70.9)			123 (61.2)	698 (71.2)		
Permanent changes	74 (6.3)	37 (15.5)	37 (3.9)			42 (20.9)	32 (3.3)		
*Compliance with Prevention and Control Measures*									
Strict, only go out for food or medicine	868 (73.5)	148 (62.2)	720 (76.4)	24.455	<0.001	127 (63.2)	741 (75.6)	14.825	0.002
Barely go out	169 (14.3)	55 (23.1)	114 (12.1)			44 (21.9)	125 (12.8)		
Was strict at beginning, but gradually relaxed	104 (8.8)	28 (11.8)	76 (8.1)			21 (10.4)	83 (8.5)		
Continue to go out as usual	40 (3.4)	7 (2.9)	33 (3.5)			9 (4.5)	31 (3.2)		
*How Long Strict Control Measures can be Tolerated*									
I cannot even endure 1–2 weeks	5 (0.4)	3 (1.3)	2 (0.2)	41.040	<0.001	1 (0.5)	4 (0.4)	33.048	<0.001
A month at maximum	47 (4.0)	9 (3.8)	38 (4.0)			9 (4.5)	38 (3.9)		
3 months maximum	247 (20.9)	74 (31.1)	173 (18.3)			69 (34.3)	178 (18.2)		
6 months maximum	327 (27.7)	80 (33.6)	247 (26.2)			59 (29.4)	268 (27.3)		
I can endure it for a long time	555 (47.0)	72 (30.3)	483 (51.2)			63 (31.3)	492 (50.2)		
*Approach to Children*									
Isolated, with no visits	390 (33.0)	82 (34.5)	308 (32.7)	0.557	0.906	65 (32.3)	325 (33.2)	3.897	0.273
Minimize visits (approx. 1/week)	238 (20.2)	47 (19.7)	191 (20.3)			41 (20.4)	197 (20.1)		
Visits as usual	156 (13.2)	33 (13.9)	123 (13.0)			19 (9.5)	137 (14.0)		
Not applicable, as living with children	397 (33.6)	76 (31.9)	321 (34.0)			76 (37.8)	321(32.8)		
*Impact of Pandemic on Health Condition **									
No effect	583 (91.8)	129 (83.2)	454 (94.6)	20.102	<0.001	96 (80.0)	487 (94.6)	27.454	<0.001
Yes	52 (8.2)	26 (16.8)	26 (5.4)			24 (20.0)	28 (5.4)		
*Exercise Maintenance during Outbreaks*									
Exercise in home	870 (73.7)	178 (74.8)	692 (73.4)	0.843	0.839	146 (72.6)	724 (73.9)	0.997	0.802
Exercise in residential area	231 (19.6)	47 (19.7)	184 (19.5)			43 (21.4)	188 (19.2)		
Exercise alone in green parks	69 (5.8)	11 (4.6)	58 (6.2)			11 (5.5)	58 (5.9)		
Exercise in a group	11 (0.9)	2 (0.8)	9 (1.0)			1 (0.5)	10 (1.0)		
None of the above	0								

Note: some items allowed single responses and others multiple. * participants reporting no condition excluded.

**Table 4 jcm-11-07275-t004:** Multivariate Analysis of Variance Assessing Correlates of Depressive and Anxious Symptoms.

	Pillai’s Trace	Wilk’s λ	F	*p*	η^2^
Sex	0.002	0.998	1.245	0.288	0.002
Age	0.011	0.989	1.589	0.123	0.005
Income Status	0.010	0.990	2.973	0.018	0.005
Hypertension	0.012	0.988	7.071	0.001	0.012
Coronary Artery Disease	0.034	0.966	20.573	<0.001	0.034
Perspective on Global COVID-19 control	0.016	0.984	3.224	0.004	0.008
*Perceptions Regarding COVID-19 Spread*					
Concerned about relatives and friends abroad	0.011	0.989	6.826	0.001	0.011
Glad China has tried to keep the outbreak under control	0.003	0.997	1.896	0.151	0.003
Worried soon second wave due to incomers	0.022	0.978	13.151	<0.001	0.022
Pandemic has spread globally, and affected economy	0.022	0.978	13.295	<0.001	0.022
Temporary issue; countries will be united in efforts so there is no need to be dejected	0.029	0.971	17.368	<0.001	0.029
Only China is my concern; not our business	0.002	0.998	1.288	0.276	0.002
Impact of Pandemic on Life	0.101	0.899	20.953	<0.001	0.051
Compliance with Prevention and Control Measures	0.026	0.974	5.253	<0.001	0.013
Impact of Pandemic on Health Condition	0.077	0.923	49.230	<0.001	0.077
How Long Strict Control Measures can be Tolerated	0.052	0.948	7.824	<0.001	0.026

## Data Availability

The dataset used for the current study is available from the Xia Liu on reasonable request.

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
