# Peer review of "Older Adults’ Attitudes Regarding COVID-19 and Associated Infection Control Measures in Shanghai and Impact on Well-Being"

_jcm, 2022, doi:10.3390/jcm11247275_

Round 1

Reviewer 1 Report

Thank you for submitting your interesting study for review. 

This study takes the opportunity to study a unique effect that occurred across the world during the COVID pandemic, and offers insight into the impact and effects of social restrictions on the older population in Shanghai, China. In that, it offers insight into this population, though, as the authors acknowledge, this may lack generalisability to western contexts. 

The background and introduction is useful and explains some of the social context of China and Shanghai and Chinese social culture relevant to the study. The method is fairly well explained except the survey resulting in Table 3 is not explained. It would be useful to know more about how the questions were devised. This was not clear to me. 

Analysis appears appropriate. Table 3 was somewhat confusing. I suggest the variables and sub variables headings needs to be in column 1 for improved clarity. also, as there is no direct comparison between depression and anxiety, it may help to split these two independent variables into 2 tables: this would make Table 3 details more accessible. 

One factor that needs explaining and incorporating into the analysis and interpretation, I would suggest, is the timing of the survey. It is not clear at what stage of social restrictions this survey was carried out: how long people had been in isolation and restriction and whether the data collection was retrospective or prospective. This is important to understand people's perceptions at the time, especially during a time of 'wellbeing paradox' as the authors suggest. Also, of course, as the authors rightly point out, the key further research would be a follow up of perceptions of the impact of the restrictions and its longer term impacts, in comparison to perceptions during the early stages of the pandemic. 

One further observation; it is not clear why details of the psychoactive prescriptions and pre-pandemic life events were explained, especially when the latter factor was not significant. Perhaps there was a point to be made that was not later explored? Also, the self report that people thought they could endure the restrictions for 3 or 6 months is fairly meaningless except as a proxy measure of their current motivation, as they had not actually endured it for that long (or so it appears from the reporting). This also is a detail not explored. 

Reviewer 2 Report

First of all, the abstract is well formulated and covers all the important parts of the article.

The introduction should be rewritten a bit to make the sentences clearer. Feel free to break some of the longer sentences in two, to make the read easier and to formulate the ideas more clearly.

The materials and methods section is well spaced and well written.

For table 2, It would be more appropriate to either replace the percentage in the Effect and No Effect columns refers to the percentages of people who suffer from the underlying condition. (I.e. the percentage of hypertensive patients who have observed an effect from the pandemic on their condition) or to specify in the thing studied is whether the person was influenced by the pandemic and that these are the variables that influenced it.

Please move the paragraphs that discuss the results in table 2-4 to the discussion section.

Figure 1 is nice, but is really not needed, especially at the end of the article.
